

# Population structure and phenotypic variation of *Sclerotinia sclerotiorum* from dry bean (*Phaseolus vulgaris*) in the United States

Zhian N. Kamvar[1], B. Sajeewa Amaradasa[1,2], Rachana Jhala[1,3], Serena McCoy[1], James R. Steadman[1] and Sydney E. Everhart[1]

[1] Department of Plant Pathology, University of Nebraska, Lincoln, NE, USA
[2] Current affiliation: Plant Pathology Department, University of Florida, Gainsville, FL, USA
[3] Current affiliation: Nebraska Center for Virology, University of Nebraska-Lincoln, Lincoln, NE, USA

Corresponding author
Sydney E. Everhart,
everhart@unl.edu

## ABSTRACT

The ascomycete pathogen *Sclerotinia sclerotiorum* is a necrotrophic pathogen on over 400 known host plants, and is the causal agent of white mold on dry bean. Currently, there are no known cultivars of dry bean with complete resistance to white mold. For more than 20 years, bean breeders have been using white mold screening nurseries (wmn) with natural populations of *S. sclerotiorum* to screen new cultivars for resistance. It is thus important to know if the genetic diversity in populations of *S. sclerotiorum* within these nurseries (a) reflect the genetic diversity of the populations in the surrounding region and (b) are stable over time. Furthermore, previous studies have investigated the correlation between mycelial compatibility groups (MCG) and multilocus haplotypes (MLH), but none have formally tested these patterns. We genotyped 366 isolates of *S. sclerotiorum* from producer fields and wmn surveyed over 10 years in 2003–2012 representing 11 states in the United States of America, Australia, France, and Mexico at 11 microsatellite loci resulting in 165 MLHs. Populations were loosely structured over space and time based on analysis of molecular variance and discriminant analysis of principal components, but not by cultivar, aggressiveness, or field source. Of all the regions tested, only Mexico ($n = 18$) shared no MLHs with any other region. Using a bipartite network-based approach, we found no evidence that the MCGs accurately represent MLHs. Our study suggests that breeders should continue to test dry bean lines in several wmn across the United States to account for both the phenotypic and genotypic variation that exists across regions.

## INTRODUCTION

*Sclerotinia sclerotiorum* (Lib.) de Bary is an ascomycete plant pathogen with a worldwide distribution (*Bolton, Thomma & Nelson, 2006*). This is a necrotrophic pathogen that is primarily homothallic (self-fertilization) and has the ability to survive for more than five years in soil using melanized survival structures called sclerotia (*Bolton,*

*Thomma & Nelson, 2006*; *Sexton, Whitten & Howlett, 2006*). It causes disease on more than 400 plant species belonging to 75 families (*Boland & Hall, 1994*) including crops of major economic importance such as sunflower (*Helianthus* spp.), soybean (*Glycine max* L.), canola (*Brassica rapa* L., *Brassica campestris* L.), and dry bean (*Phaseolus vulgaris* L.) (*Bolton, Thomma & Nelson, 2006*).

On dry bean, *Sclerotinia sclerotiorum* is the causal agent of white mold, a devastating disease that can be yield-limiting in temperate climates (*Steadman, 1983*). All above-ground tissues (flowers, stems, leaves, pods) are susceptible to infection, first appearing as wet lesions with white mycelial tufts, and then bleaching as the tissue senesces (*Steadman, 1983*; *Bolton, Thomma & Nelson, 2006*). For many years, white mold has been the most serious dry bean disease in the Northwestern United States (*Otto-Hanson et al., 2011*; *Knodel et al., 2012*, *2015*, *2016*). The impact of white mold on the dry bean industry in the Northwestern United States alone has been estimated at a loss of 140 kg/ha with just 10% disease incidence (*Ramasubramaniam, del Río Mendoza & Bradley, 2008*).

Currently, there are no commercially available resistant cultivars of dry bean (*Otto-Hanson et al., 2011*). Organized breeding efforts have used a common-garden approach with white mold screening nurseries (wmn) in dry bean production areas across the United States with additional sites in Australia, France, and Mexico (*Steadman, Eskridge & Powers, 2003*; *Steadman, Otto-Hanson & Powers, 2004*, *2005*; *Steadman, Otto-Hanson & Breathnach, 2006*; *Otto-Hanson & Steadman, 2007*, *2008*; *McCoy & Steadman, 2009*). These wmn use no chemical or cultural treatments against *S. sclerotiorum* and employ standardized protocols for screening new cultivars for resistance to white mold (*Steadman, Eskridge & Powers, 2003*; *Otto-Hanson et al., 2011*). These protocols included three established cultivars used for comparison in the trials: Beryl (great northern bean, susceptible), Bunsi (a.k.a. Ex Rico, navy bean, low susceptibility), and G122 (cranberry bean, partial resistance) (*Tu & Beversdorf, 1982*; *Steadman, Otto-Hanson & Powers, 2005*; *Otto-Hanson et al., 2011*). It was previously shown that aggressiveness (the severity of disease symptoms on the host) is significantly different across white mold screening nursery sites in separate geographic regions (*Otto-Hanson et al., 2011*). The genetic structure and mode of reproduction in these populations, however, is currently unknown.

Understanding genetic relationships and reproduction behavior of *S. sclerotiorum* populations is beneficial for breeders seeking to develop new resistant cultivars for worldwide deployment (*Milgroom, 1996*; *McDonald & Linde, 2002*). In particular, genetically diverse populations with high rates of sexual reproduction are more likely to overcome host resistance. Most populations of *S. sclerotiorum* are predominantly clonal with low genetic diversity and have a large degree of population fragmentation (*Kohli et al., 1995*; *Cubeta et al., 1997*; *Kohli & Kohn, 1998*; *Carbone & Kohn, 2001*; *Ekins et al., 2011*; *Attanayake et al., 2012*). Some studies, however have found populations that show signatures of sexual reproduction (*Atallah et al., 2004*; *Sexton & Howlett, 2004*; *Attanayake et al., 2013*; *Aldrich-Wolfe, Travers & Nelson, 2015*).

Nearly all population genetic studies of *S. sclerotiorum* employ a macroscopic assay to determine mycelial compatibility, the ability for fungal hyphae from different colonies to

appear to grow together without forming a barrier of dead cells between them (known as a barrage line, Fig. S1B) (*Leslie, 1993*; *Sirjusingh & Kohn, 2001*). Mycelial compatibility has been used as a proxy for vegetative compatibility, a fungal trait controlled by several independent genes that mitigate the ability for two hyphae to fuse and grow as a single unit (Fig. S1A) (*Leslie, 1993*; *Schafer & Kohn, 2006*). Because of the genetic connection to vegetative compatibility, two isolates that are mycelially compatible were considered clones (*Leslie, 1993*); but correlation with genetic markers, such as microsatellites, have shown inconsistent results (*Ford et al., 1995*; *Micali & Smith, 2003*; *Jo et al., 2008*; *Attanayake et al., 2012*; *Papaioannou & Typas, 2014*; *Lehner et al., 2017*). Thus, the relationship between mycelial compatibility groups (MCGs) and clonal genotypes remains unclear.

In the present study, we analyze and characterize the genetic and phenotypic diversity of 366 *S. sclerotiorum* isolates collected between 2003 and 2012 from dry bean cultivars among different geographic locations in the Australia, France, Mexico, and the United States. We wanted to know if the *S. sclerotiorum* populations from wmn were representative of the producer fields within the same region. As these nurseries were not treated with any chemical or cultural control of white mold, we hypothesized that these nurseries would represent the natural population of *S. sclerotiorum*. Furthermore, we wanted to investigate the potential effect of cultivar on genetic diversity of the pathogen by assessing three dry bean cultivars with different levels of resistance, Beryl (great northern bean, susceptible), Bunsi (navy bean, low susceptibility), and G122 (cranberry bean, partial resistance) (*Otto-Hanson et al., 2011*). We additionally wanted to determine categorical or phenotypic variables that best predicted genetic structure and if there was correlation between multilocus haplotype (MLH) and MCG. Knowing what variables predict genetic structure can help direct breeding efforts. By investigating these aims, we will effectively describe the population structure of *S. sclerotiorum* in the United States and make available our database of isolates for use in future dry bean breeding efforts.

## MATERIALS AND METHODS

### Isolate collection

Several (156) of the isolates used for this study were collected as reported in previous studies using the same methods (*Otto-Hanson et al., 2011*). Broadly, isolates were collected from two sources: wmn or producer fields. wmn were 5 × 10 m in size and maintained without application of fungicides to observe natural incidence of white mold. The early nursery plots were incorporated with a basal dressing of N:P:K = 1:3:2 and side dressing of 0:3:2 during the growing season (*Steadman, Eskridge & Powers, 2003*).

Sampling was carried out by collecting sclerotia from diseased tissue in zigzag transects across field plots. Because sampling depended on disease incidence, the number of samples isolated varied from year to year. Although the nursery locations were the same over sampling years, sampling plots within a location varied for sampling years.

Sclerotia of *S. sclerotiorum* were collected over several years from grower fields and/or wmn in 11 states of the Australia, France, Mexico, and the United States (Table S1). After collection, sclerotia were stored in Petri plates lined with filter paper, then stored at 20 °F or −4 °C. Sclerotia were surface-sterilized with 50% Clorox bleach (at least 6% NaOCl, The Clorox Company, Oakland, CA, US) solution for 3 min, and double rinsed with ddH$_2$O for 3 min. The sterilized sclerotia were then placed on water agar plates (16 g of Bacto agar per liter of ddH$_2$O; BD Diagnostic Systems, Sparks, MD, US), with four to five sclerotia of each isolate separated on each plate and stored on the counter top at room temperature for five to six days. An 8 mm plug from a 5- or 6-day-old culture was transferred from the advancing margin of the mycelia onto a plate of Difco potato dextrose agar (PDA at 39 g/l of ddH$_2$O) (*Otto-Hanson et al., 2011*). In combination with the 156 isolates described previously, we collected 210 isolates for a total of 366 isolates (*Otto-Hanson et al., 2011*).

## Mycelial compatibility

Mycelial compatibility groups was determined as described previously through co-culturing pairs of 2-day-old isolates 2.5 cm apart on Diana Sermons (DS) Medium (Fig. S1) (*Cubeta, Sermons & Cody, 2001*). Incompatibility of different MCGs resulted in formation of a barrage line accompanied by formation of sclerotia on either side of the barrage line, indicating the limits of mycelial growth (*Kohn, Carbone & Anderson, 1990*; *Leslie, 1993*; *Otto-Hanson et al., 2011*). Isolates were compared in a pairwise manner for each site and then representatives among sites were compared to determine MCGs by scoring compatible and incompatible interactions (*Otto-Hanson et al., 2011*). No MCGs were compatible with any other MCG.

## Aggressiveness

Aggressiveness of each isolate was assessed using a straw test as described in *Otto-Hanson et al. (2011)* that rated necrotic lesion size (*Petzoldt & Dickson, 1996*; *Teran et al., 2006*). Briefly, the straw test uses 21-day-old G122 plants as the host in a greenhouse setting. Clear drinking straws cut to 2.5 cm and heat sealed were used to place two mycelial plugs of inoculum on the host plant after removing plant growth beyond 2.5 cm above the fourth node. Measurements of the necrotic lesion were taken eight days later using the Modified Petzoldt and Dickson scale of 1–9, where 1 is no disease and 9 is plant death (*Petzoldt & Dickson, 1996*; *Teran et al., 2006*).

## Microsatellite genotyping

Prior to DNA extraction, isolates were grown on PDA and plugs were subsequently transferred to potato dextrose broth where they were grown until there was significant mycelial growth, but before the mycelial mat became solidified (four to five days). Each mycelial mat was collected in a filtered Büchner funnel, agar plugs removed, lyophilized and pulverized manually in Whirl-pak® HDPE sampling bags (Sigma-Aldrich, St. Louis, MO, US). Lyophilized mycelia was then stored in microcentrifuge tubes at −20 °C until needed for DNA extraction. DNA from 25 mg of pulverized mycelia was

purified using a phenol–chloroform extraction method followed by alcohol precipitation and evaporation, suspending the DNA in 200 μl TE (*Sambrook, Fritsch & Maniatis, 1989*). Suspended DNA was stored at 4 °C until genotyping.

We genotyped each *S. sclerotiorum* isolate using 16 microsatellite primer pairs developed previously (*Sirjusingh & Kohn, 2001*). PCR was carried out as described previously, using primers labeled with FAM fluorophore. Resulting amplicons were first resolved in a 1.5% agarose gel stained with ethidium bromide to ensure product was within the expected size range prior to capillary electrophoresis. Capillary electrophoresis (fragment analysis) of amplicons, with size standard GeneScan™ 500 LIZ®, was performed using an ABI 3730 genetic analyzer (Life Technologies Corporation, Carlsbad, CA, US) at the Michigan State University Genomic Sequencing Center (East Lansing, MI, US). Alleles were scored using PeakScanner version 1.0 (Life Technologies Corporation, Carlsbad, CA, US) and recorded manually in a spreadsheet.

## Data processing and analysis

All data processing and analyses were performed in a Rocker "verse" project container running R version 3.4.2 (*Boettiger & Eddelbuettel, 2017*; *R Core Team, 2017*). These analyses were rendered as dynamic documents with the R packages knitr (version 1.17) and ezknitr (version 0.6) and are openly available and reproducible at https://github.com/everhartlab/sclerotinia-366/ (*Attali, 2016*; *Xie, 2017*). Of the 16 microsatellite loci genotyped, five included compound repeats, which made it challenging to accurately/confidently bin alleles into fragment sizes expected for each locus based on the described repeat motif. Loci with compound repeats were removed for the reported statistics. To ensure the integrity of the results we additionally processed these loci and included them in concurrent analyses. We assessed the power of our 11 markers by generating a genotype accumulation curve in the R package *poppr* version 2.5.0, looking for evidence of saturation, which would indicate that loci were sufficiently sampled to adequately represent the full set of haplotypes (*Arnaud-Hanod et al., 2007*; *Kamvar, Brooks & Grünwald, 2015*). We additionally assessed within-locus allelic diversity by measuring Nei's gene diversity ($h$) (*Nei, 1978*) and allelic evenness ($E_5$) (*Pielou, 1975*; *Grünwald et al., 2003*). To avoid including isolates potentially collected from the same plant (which increases the probability of collecting sclerotia from the same point of infection more than once), data were clone-corrected on a hierarchy of Region/Source/Host/Year—meaning that duplicated genotypes were reduced to a single observation when they were collected in the same year from the same host cultivar located in the same source field (wmn or producer)—for subsequent analysis. We assessed haplotype diversity by calculating Stoddart and Taylor's index ($G$) (*Stoddart & Taylor, 1988*), Shannon's index ($H$) (*Shannon, 1948*), Simpson's index ($\lambda$) (*Simpson, 1949*), evenness ($E_5$), and the expected number of multilocus haplotypes via rarefaction (*eMLH*) with 10 samples (*Hurlbert, 1971*; *Heck, van Belle & Simberloff, 1975*; *Pielou, 1975*; *Grünwald et al., 2003*). If all haplotypes in are equally abundant, both G and $e^H$ (the exponentiation of H) are expected to be equal to the number of haplotypes, $\lambda$ and $E_5$ are expected to equal one, and *eMLH* is expected to be at its maximum value (in this case, 10) (*Grünwald et al., 2003*). To assess the potential for

random mating, we tested for linkage disequilibrium with the index of association, $I_A$ and its standardized version, $\bar{r}_d$ using 999 permutations (*Brown, Feldman & Nevo, 1980*; *Smith et al., 1993*; *Agapow & Burt, 2001*). Both haplotype diversity and linkage disequilibrium were calculated in *poppr* (*Kamvar, Tabima & Grünwald, 2014*).

## Assessing importance of variables

### Distance-based redundancy analysis

A distance-based redundancy analysis (dbRDA) (*Legendre & Anderson, 1999*) was performed with the function `capscale()` in the *vegan* package version 2.4.4 (*Oksanen et al., 2017*). This method uses constrained ordinations on a distance matrix representing the response variable to delineate relative contribution of any number of independent explanatory variables. We used this method to delineate the phenotypic (Aggressiveness, MCG), geographic (Region, Host, Location), and temporal (Year) components in predicting genetic composition of the populations. The distance matrix we used as the response variable was generated using Bruvo's genetic distance from clone-corrected data (procedure described above) as implemented in *poppr*, which employed a stepwise mutation model for microsatellite data (*Bruvo et al., 2004*; *Kamvar, Tabima & Grünwald, 2014*). Because aggressiveness measures differed between isolates that were reduced to a single observation during clone-correction, aggressiveness was first averaged across clone-corrected isolates. To identify explanatory variable(s) correlated with genetic variation, a forward–backward selection process was applied with the vegan function `ordistep()`. An analysis of variance (ANOVA) was then performed to test for significance of the reduced model and marginal effects using 999 permutations. The `varpart()` function of vegan was used to determine variation partitioning of explanatory variables.

### Aggressiveness assessment

We used ANOVA to assess if aggressiveness (determined via straw test on a scale of 1–9 as described above) was significantly different with respect to Region, MCG, or MLH. To minimize complications due to small sample sizes, we chose the top 10 MCGs, representing 56.5% of the isolates collected, the 10 most abundant MLHs representing 26.7% of the isolates, and populations with more than five isolates. If ANOVA results were significantly different at $\alpha = 0.05$, pairwise differences were assessed using Tukey's HSD test ($\alpha = 0.05$) using the `HSD.test()` function in the package *agricolae* version 1.2.8 (*Mendiburu & Simon, 2015*).

### Correlating MLH with MCGs

We wanted to assess if there was correlation between MLHs and MCGs. This was performed using a network-based approach where both MLHs and MCGs were considered nodes and the number of isolates in which they were found together was the strength of the connection between an MLH and MCG node. The network-based approach allowed us to assess the associations between MLHs and MCGs. To construct the network, a contingency table was created with MLHs and MCGs and converted to a directed and weighted edgelist where each edge represented a connection from an MCG to

an MLH, weighted by the number of samples shared in the connection. This was then converted to a bipartite graph where top nodes represented MLHs and bottom nodes represented MCGs. To identify clusters of MLHs and MCGs within the network, we used the cluster walktrap community detection algorithm as implemented in the `cluster_walktrap()` function in *igraph* version 1.1.2 (*Csardi & Nepusz, 2006*; *Pons & Latapy, 2006*). This algorithm attempts to define clusters of nodes by starting at a random node and performing short, random "walks" along the edges between nodes, assuming that these walks would stay within clusters. For this analysis, we set the number of steps within a walk to four and allowed the algorithm to use the edge weights in determining the path. All of the resulting communities that had fewer than 10 members were then consolidated into one. Community definitions were used to assess the average genetic distance (as defined by Bruvo's distance) within members of the community (*Bruvo et al., 2004*).

## Genetic diversity

### Population differentiation

We used analysis of molecular variance (AMOVA) with Bruvo's genetic distance in *poppr* to test for differentiation between populations in wmn and producer fields from the same region and collected in the same year (*Excoffier, Smouse & Quattro, 1992*; *Bruvo et al., 2004*; *Kamvar, Tabima & Grünwald, 2014*). To identify Regions with greater differentiation, we used discriminant analysis of principal components (DAPC) as implemented in *adegenet* version 2.1.0, assessing the per-sample posterior group assignment probability (*Jombart, 2008*). This method decomposes the genetic data into principal components, and then uses a subset of these as the inputs for discriminant analysis, which attempts to minimize within-group variation and maximize among-group variation (*Jombart, Devillard & Balloux, 2010*). To avoid over-fitting data, the optimal number of principal components was selected by using the *adegenet* function `xvalDapc()`. This function implements a cross-validation procedure to iterate over an increasing number of principal components on a subset (90%) of the data, trying to find the minimum number of principal components that maximizes the rate of successful group reassignment. To assess if cultivar had an influence on genetic diversity between wmn, we first subset the clone-corrected data to contain only samples from wmn and from the cultivars Beryl, Bunsi, and G122 and tested differentiation using AMOVA and DAPC as described above. We additionally assessed population stability over time by calculating DAPC over the combined groups of Region and Year as described above.

### Analysis of shared MLH

We wanted to evaluate patterns of connectivity between shared MLH across geographic regions. We first tabulated the MLH shared between at least two populations (defined as states or countries) with the *poppr* function `mlg.crosspop()` (*Kamvar, Tabima & Grünwald, 2014*). From these data, we constructed a graph with populations as nodes and shared haplotypes as edges (connections) between nodes using the R packages *igraph*

(*Csardi & Nepusz, 2006*), *dplyr* version 0.7.4 (*Wickham et al., 2017*), and *purrr* version 0.2.4 (*Henry & Wickham, 2017*). Each node was weighted by the fraction of shared MLHs. Each edge represented a single MLH, but because a single MLH could be present in more than one population, that MLH would have a number of edges equivalent to the total number of possible connections, calculated as $(n^*(n-1))/2$ edges where $n$ represents the number of populations crossed. Edges were weighted by $1 - P_{sex}$, where $P_{sex}$ is the probability of encountering the same haplotype via two independent meiotic events (*Parks & Werth, 1993*; *Arnaud-Hanod et al., 2007*). This weighting scheme would thus strengthen the connection of edges that represented genotypes with a low probability of being produced via sexual reproduction. We then identified communities (among the Regions) in the graph using the `cluster_optimal()` function from *igraph* (*Csardi & Nepusz, 2006*). The graph was plotted using the R packages *ggplot2* version 2.2.1 (*Wickham, 2009*) and *ggraph* 1.0.0 (*Pedersen, 2017*). To ensure that we captured the same community signal, we additionally performed this analysis including the five polymorphic markers described above.

## RESULTS

A total of 366 isolates were collected from 2003 to 2012 (except 2006 and 2011) from diseased dry bean plants in 11 states in the United States as well as Australia, France, and Mexico (Table S1). With the 11 loci used in the analyses (Table 1), we observed a total of 165 MLHs (215 with 16 loci). These 11 loci are located on seven chromosomes in the *S. sclerotiorum* genome with a minimum distance of 55 Kbp between two loci on the same chromosome. Over 50% of the isolates came from four states, MI (62), ND (60), WA (59), NE (47). Four regions had fewer than 10 isolates, Australia (6), WI (2), NY (1), ID (1). We observed 87 MCGs, the most abundant of which ("MCG 5") was represented by 73 isolates over 37 MLHs (Figs. 1A and 1C).

The number of observed alleles per locus ranged from 2 to 10 with an average of 6.27 (Table 1). Locus 20-3, which contained only two alleles, showed low values of both $h$ (0.0533) and $E_5$ (0.42), indicating that there was one dominant allele present. Analysis of the haplotype accumulation curve showed no clear plateau for 11 or 16 loci (see section on "Loading Data and Setting Strata" in the MLG-distribution.md file in the supplemental files, https://github.com/everhartlab/sclerotinia-366/blob/master/results/MLG-distribution.md#loadingdata-and-setting-strata; *Kamvar et al., 2017*), indicating that we would likely obtain more MLH if we were to genotype more loci.

After clone-correction on the hierarchy of Region/Source/Host/Year, a total of 48 isolates were removed from the data set, resulting in 318 isolates representing 165 MLHs that were used in subsequent analyses (Table 2). The results showed that, in terms of genotypic diversity ($H$, $G$, and $\lambda$), WA was the most diverse population with both $G$ (54.3) and $e^H$ (55.3) being close to the observed number of MLHs (56). This indicated that there are few duplicated genotypes in WA (Table 2). A more useful metric to compare populations, however, is $E_5$, which scales from 0 to 1, where 1 indicates all unique genotypes (*Grünwald et al., 2003*). Evaluating by $E_5$ shows that both MI and NE exhibit lower than average values, indicating that there are

**Table 1 Allelic diversity on full data set at loci used in this study.**

| Locus | Range | Repeat motif | Number of alleles | $h$ | $E_5$ |
|-------|-------|--------------|-------------------|-----|-------|
| 5-2 | 318–324 | (GT) | 4 | 0.45 | 0.62 |
| 6-2 | 483–495 | (TTTTTC)(TTTTTG) (TTTTTC) | 3 | 0.64 | 0.95 |
| 7-2 | 158–174 | (GA) | 7 | 0.73 | 0.76 |
| 8-3 | 244–270 | (CA) | 7 | 0.74 | 0.79 |
| 9-2 | 360–382 | (CA)(CT) | 9 | 0.35 | 0.41 |
| 12-2 | 214–222 | (CA) | 5 | 0.58 | 0.78 |
| 17-3 | 342–363 | (TTA) | 7 | 0.55 | 0.53 |
| 20-3 | 280–282 | (GT)GG(GT) | 2 | 0.05 | 0.42 |
| 55-4 | 153–216 | (TACA) | 10 | 0.72 | 0.66 |
| 110-4 | 370–386 | (TATG) | 5 | 0.76 | 0.91 |
| 114-4 | 339–416 | (TAGA) | 10 | 0.83 | 0.80 |

Note:
   $h$ = Nei's 1978 gene diversity, $E_5$ = Evenness. Average $h$ = 0.583, average $E_5$ = 0.693, average no. alleles = 6.27.

over-represented genotypes in the populations (Table 2). When we look at Mexico, we observed that it had relatively high values of $E_5$ and genotypic diversity, but low richness, as measured by $eMLG$. Moreover, Mexico had the lowest value for $h$, which is a measure of allelic diversity. Nearly all populations showed evidence of linkage (Table 2), which serves as evidence for clonal reproduction or other forms of non-random mating. The only exceptions were CA ($P = 0.043$) and Australia ($P = 0.052$). Both of these populations showed only moderate significance with $\bar{r}_d$ values of 0.03 and 0.12, respectively.

## Variable assessment

### Variable contributions

The forward–backward selection process of the dbRDA models on clone-corrected data revealed Year, Region, Host, and MCG to be the optimal variables for the reduced model, accounting for 45% of the total variation. ANOVA showed that the reduced model was significant with an adjusted $R^2$ of 0.0675 ($P = 0.001$). Assessment of the marginal effects showed that all variables significantly explained genetic variation ($P \leq 0.007$). We found that there was multicollinearity when MCG was combined with any other variable, so repeated the analysis, dropping MCG from the list of potential predictors. From these results, Year, Region, Host, and Aggressiveness were found to be optimal, accounting for 17.6% of the total variation. ANOVA revealed significant effects with an adjusted $R^2$ of 0.0325 ($P = 0.001$). While the marginal effect assessment revealed that Year, Region, and Host significantly explained variation at $P = 0.001$, and Aggressiveness significantly explained variation at $P = 0.039$. Much of the variation appeared to be driven by isolates from Mexico and 2005 (Fig. 2). Variance partitioning of the independent variables without MCG indicated aggressiveness to be the least influential factor with 0.1% contributing to explaining the variation of molecular data, whereas the combination of variables accounted for 3.3%.

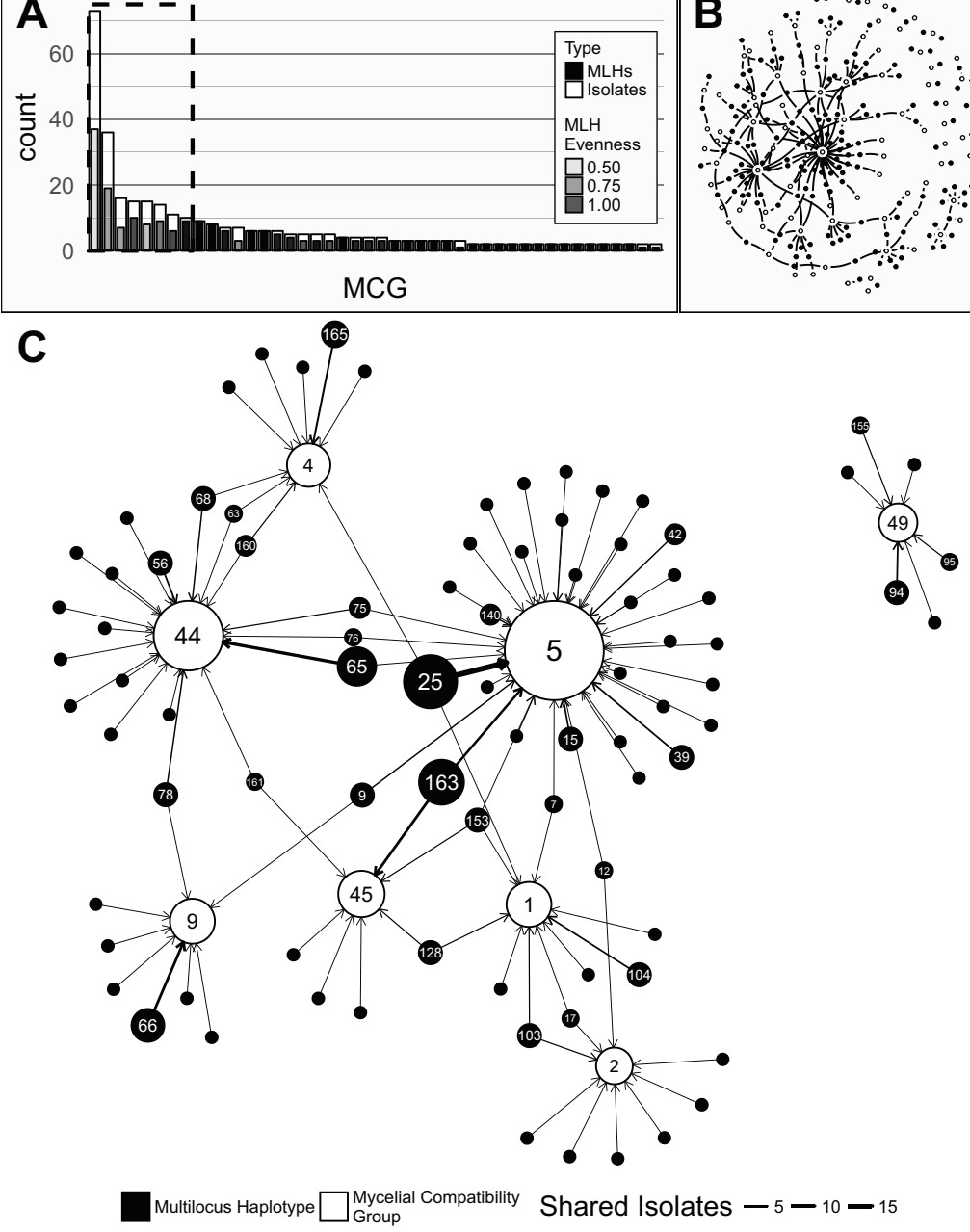

**Figure 1** **Associations between mycelial compatibility groups (MCGs) and multilocus haplotypes (MLH).** (A) Barplot of MCG abundance in descending order. Singletons (46) were truncated, leaving 41 MCGs. White bars represent sample counts and grey bars represent counts of unique MLH. The transparency of the bars represent the evenness of the distribution of the MLHs within a given MCG. A dashed box surrounds the eight most common MCGs representing >51% of the data. (B) Full graph-representation of the relationship between MCGs (open circles) and MLHs (filled circles). Details in Fig. S3. (C) A subset of (B) representing the eight most common MCGs and their associated MLHs (dashed box in (A). Filled nodes (circles) represent MLHs and open nodes represent MCGs. Node area scaled to the number of samples represented (1–73). Numbers inside nodes are the MLH/MCG label (if *n* > 1). Edges (arrows) point from MLH to MCG where the weight (thickness) of the edge represents the number of shared isolates (1–19). Edges extending from MLHs displayed to other MCGs are not shown.

**Table 2 Genotypic diversity and linkage disequilibrium summary for geographic populations arranged by abundance after clone-correction by a hierarchy of Region/Source/Host/Year.**

| Pop | N | eMLH | H | G | λ | $E_5$ | h | $\bar{r}_d$ |
|---|---|---|---|---|---|---|---|---|
| WA | 58 (56) | 9.95 (0.23) | 4.0 | 54.3 | 0.98 | 0.98 | 0.60 | 0.07* |
| MI | 58 (43) | 9.3 (0.79) | 3.6 | 29.0 | 0.97 | 0.78 | 0.54 | 0.14* |
| ND | 41 (35) | 9.44 (0.73) | 3.5 | 25.9 | 0.96 | 0.82 | 0.54 | 0.1* |
| NE | 37 (28) | 8.93 (0.94) | 3.2 | 17.8 | 0.94 | 0.75 | 0.55 | 0.25* |
| CO | 34 (28) | 9.46 (0.67) | 3.3 | 24.1 | 0.96 | 0.92 | 0.56 | 0.27* |
| France | 21 (14) | 8.5 (0.85) | 2.6 | 12.6 | 0.92 | 0.95 | 0.48 | 0.11* |
| CA | 18 (15) | 9.12 (0.72) | 2.7 | 13.5 | 0.93 | 0.94 | 0.51 | 0.03 |
| OR | 17 (13) | 8.52 (0.85) | 2.5 | 10.7 | 0.91 | 0.89 | 0.47 | 0.1* |
| Mexico | 15 (9) | 7.1 (0.85) | 2.1 | 7.3 | 0.86 | 0.89 | 0.28 | 0.37* |
| MN | 9 (7) | 7 (0) | 1.9 | 6.2 | 0.84 | 0.93 | 0.47 | 0.19* |
| Australia | 6 (6) | 6 (0) | 1.8 | 6.0 | 0.83 | 1.00 | 0.48 | 0.12 |
| WI | 2 (2) | 2 (0) | 0.7 | 2.0 | 0.50 | 1.00 | 0.27 | – |
| NY | 1 (1) | 1 (0) | 0.0 | 1.0 | 0.00 | NaN | NaN | – |
| ID | 1 (1) | 1 (0) | 0.0 | 1.0 | 0.00 | NaN | NaN | – |

**Note:**
Pop, Population; N, number of individuals (number of MLH in parentheses); eMLH, expected number of MLHs based on rarefaction at 10 individuals (standard error in parentheses); H, Shannon–Weiner Index; G, Stoddardt and Taylor's Index; λ, Simpson's Index; h, *Nei's (1978)* gene diversity; $E_5$, Evenness; $\bar{r}_d$, standardized index of association. An asterisk indicates a significant value of $\bar{r}_d$ after 999 permutations, $P \leq 0.001$.

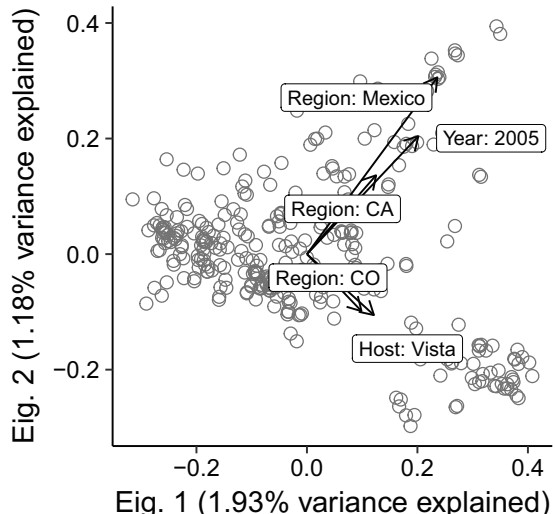

**Figure 2 Biplot showing five most influential explanatory variables (arrows) overlaid on the first two eigenvectors of distance based redundancy analysis of 318 *S. sclerotiorum* multilocus haplotypes.** The length of the arrows are directly proportional to the strength of the correlation between explanatory and molecular variables. Open circles represent the 318 clone-corrected haplotypes in ordination space.

### Aggressiveness

Aggressiveness of the isolates ranged from 1.4 to 7.9 with a mean of 5.02 and median of 4.85. The group mean averages were 4.88, 5.13, and 5.19 for Region, MCG, and MLH,

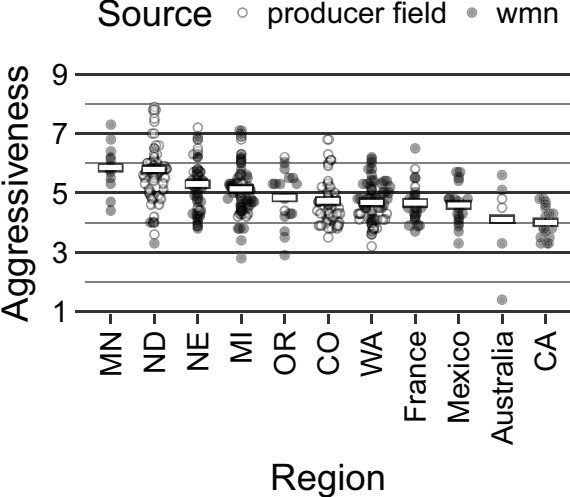

**Figure 3 Strip plot of aggressiveness by population arranged in descending order of mean aggressiveness for all populations with N > 5.** White bars represent mean value. Circles represent individual isolates where filled circles are isolates from white mold screening nurseries (wmn) and open circles are isolates from producer fields.

respectively. A strip plot showing the distribution of severity across these three variables simultaneously can be seen in Fig. S2. Our assessment of aggressiveness in association with Region showed a significant effect ($P < 1.00e^{-4}$), with means that ranged from 5.8 (MN) to 4.0 (CA) (Fig. 3; Table S2). MCGs also showed a significant effect ($P < 0.001$), with means that ranged from 6.0 ("MCG 44") to 4.6 ("MCG 49;" Table S3). We additionally found a significant effect for MLHs ($P < 0.001$), with means that ranged from 6.0 ("MLH 78") to 4.3 ("MLH 140") (Table S4).

*Correlation of MLH and MCGs*

In our analysis, we found 165 MLHs with 70 singletons and 87 MCGs with 43 singletons (Figs. 1A and 1B) where the eight most abundant MCGs represented >51% of the data over 11 Regions, and all years except for 2012. Our network-based approach to correlating MLHs with MCGs revealed a large and complex network (Fig. 1; Table 3). Community analysis showed 51 communities, 15 of which consisted of a single MLH unconnected with any other community indicating that just 9.09% of the 165 MLHs are unable to cross with any other MLH in this data set (Fig. S3). The three communities with the most members contained 8 of the 10 most abundant MCGs. Comparing these communities with Bruvo's genetic distance showed an average distance of 0.451 among communities and an average distance of 0.437 within communities, which were not significantly different. When we assessed the number of times two different MLHs that are in the same MCG, considering these as potential heterothallic pairings that could result in sexual recombination, we found an average of 14.3 potential heterothallic parings per MLH. Representing just four isolates, "MLH 75" had 57 neighbors that shared the same MCG (Fig. 1; Fig. S3). Overall, there was no clear pattern to the association between MLH and MCGs.

**Table 3 The five most abundant multilocus haplotypes (MLH) with the probability of second encounter ($P_{\text{sex}}$), mycelial compatibility groups (MCG), and Regions with sample sizes in parentheses.**

| MLH | $P_{\text{sex}}$ | MCG | Region |
| --- | --- | --- | --- |
| 25 | 0.016824 | 5 | ND (15), CO (2), MI (2) |
| | | 13 | ND (3) |
| | | 60 | ND (2), WA (1) |
| | | 1 | NE (1) |
| | | 4 | MI (1) |
| 163 | 0.049932 | 45 | CO (5), ND (2), NE (1) |
| | | 5 | MI (7) |
| 65 | 0.000071 | 44 | NE (10) |
| | | 5 | MI (1) |
| 140 | 0.000155 | 8 | CO (5) |
| | | 5 | MI (3) |
| | | 20 | MI (2) |
| 66 | 0.000016 | 9 | NE (4), CO (2), MI (2) |

## Structure of shared MLH

The most abundant MLH was represented by 27 isolates (Table 3) from five Regions (NE, MI, WA, CO, and ND). Within Regions, haplotypes were relatively evenly distributed with moderate to high diversity (Table 2). Of the 165 MLHs, 76 (46%) were found in at least two Regions, except those found in WI (2), ID (1), and Mexico (18) (Fig. 4).

We had performed an analysis on a network where the connections represented shared MLHs across populations, weighted by $1 - P_{\text{sex}}$ (Fig. 4; Table 3). Community analysis of the MLHs shared between populations revealed four communities with a modularity of 0.17: A coastal community (CA, OR, WA, and NY), a Midwest community (CO, ND, NE, MI), and an international community (Australia, France, MN). Although analysis with 16 loci resulted in the removal of the NY node because it no longer shared a haplotype with OR, the same overall community structure was present with a modularity of 0.2 (Fig. S4). Relative to the United States, the international community appears to be driven by MLH 4, which is shared between all three populations and has a $P_{\text{sex}}$ value of $2.87\text{e}^{-5}$, in contrast to the abundant MLH 25, which has a $P_{\text{sex}}$ value of 0.0168.

## Population differentiation

### Analysis of molecular variance

The AMOVA for clone-corrected haplotypes over the hierarchy of Region, Source, and Year showed significant variation between Regions and Years, but no significant variation between wmn and producer fields (Table 4). In contrast, when we compared the three cultivars, Beryl, Bunsi, and G122, we found no significant differentiation (see section on "Host Differentiation" in the wmn-differentiation.md file in the supplemental files, https://github.com/everhartlab/sclerotinia-366/blob/master/results/wmn-differentiation. md#hostdifferentiation; *Kamvar et al., 2017*).

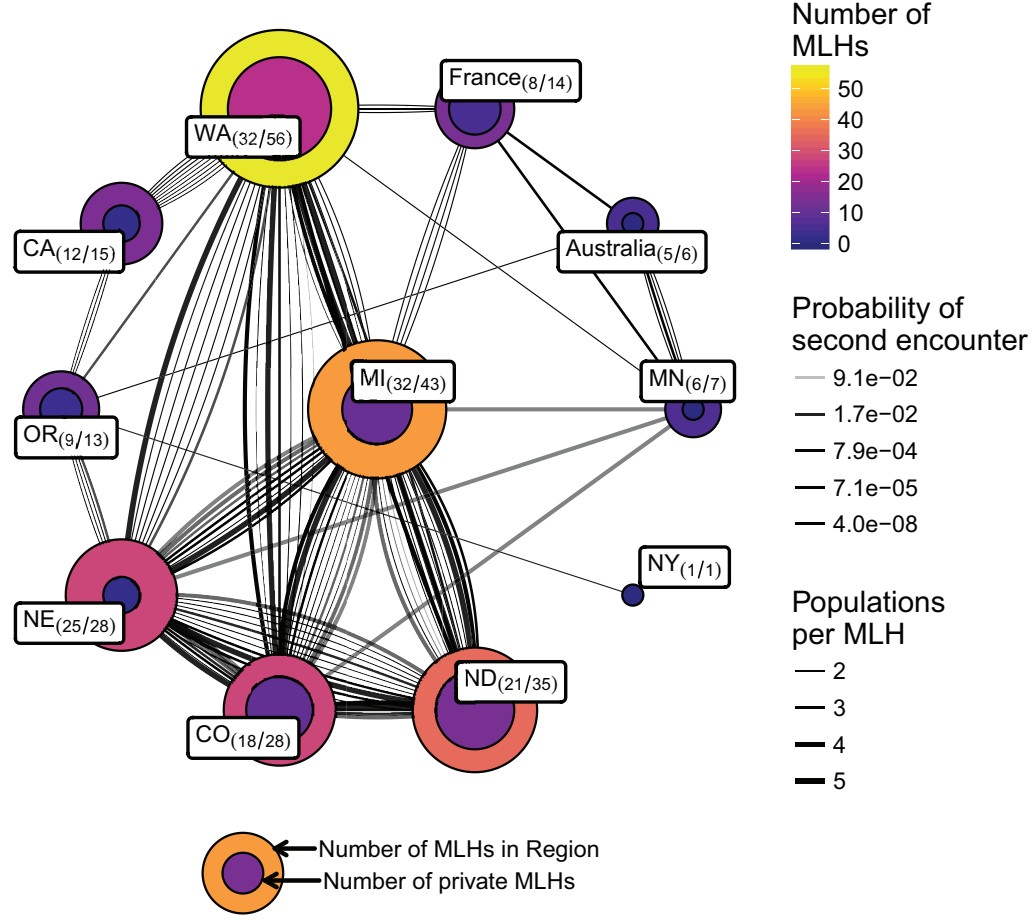

**Figure 4 Network of populations (nodes/circles) and their shared multilocus haplotypes (MLH) (edges/lines) genotyped over 11 loci.** Each node is labeled with **name (number of MLHs shared/ number of MLHs total)**. The shade and area of the nodes are proportional to the number of unique MLHs within the node and the inner nodes are proportional to the number of private MLHs to the region (bottom legend). Each edge represents a single MLH where its thickness represents the number of populations that share the MLH and the shade represents the value of $P_{sex}$, or the probability of encountering that MLH from two independent meiotic events.

**Table 4 Comparison of populations in the white mold screening nurseries (wmn) and producer fields using an analysis of molecular variance (AMOVA) on Bruvo's genetic distance showing no apparent differentiation between wmn and other sources.**

| Hierarchy | d.f. | S.S. | % Variation | $\Phi$ Statistic | P |
|---|---|---|---|---|---|
| Between Region | 13 | 10.19 | 8.45 | **0.0845** | 0.031 |
| Between Source within Region | 8 | 2.74 | −2.29 | −0.0250 | 0.497 |
| Between Year within Source | 22 | 9.37 | 16.28 | **0.173** | 0.001 |
| Within Year | 274 | 47.30 | 77.56 | **0.224** | 0.001 |

**Note:**
The hierarchy was constructed as Source/Region where source is defined as belonging to a wmn or producer field. Bold $\Phi$ values indicate significant difference ($P < 0.05$). S.S., Sum of Squares; d.f., degrees of freedom.

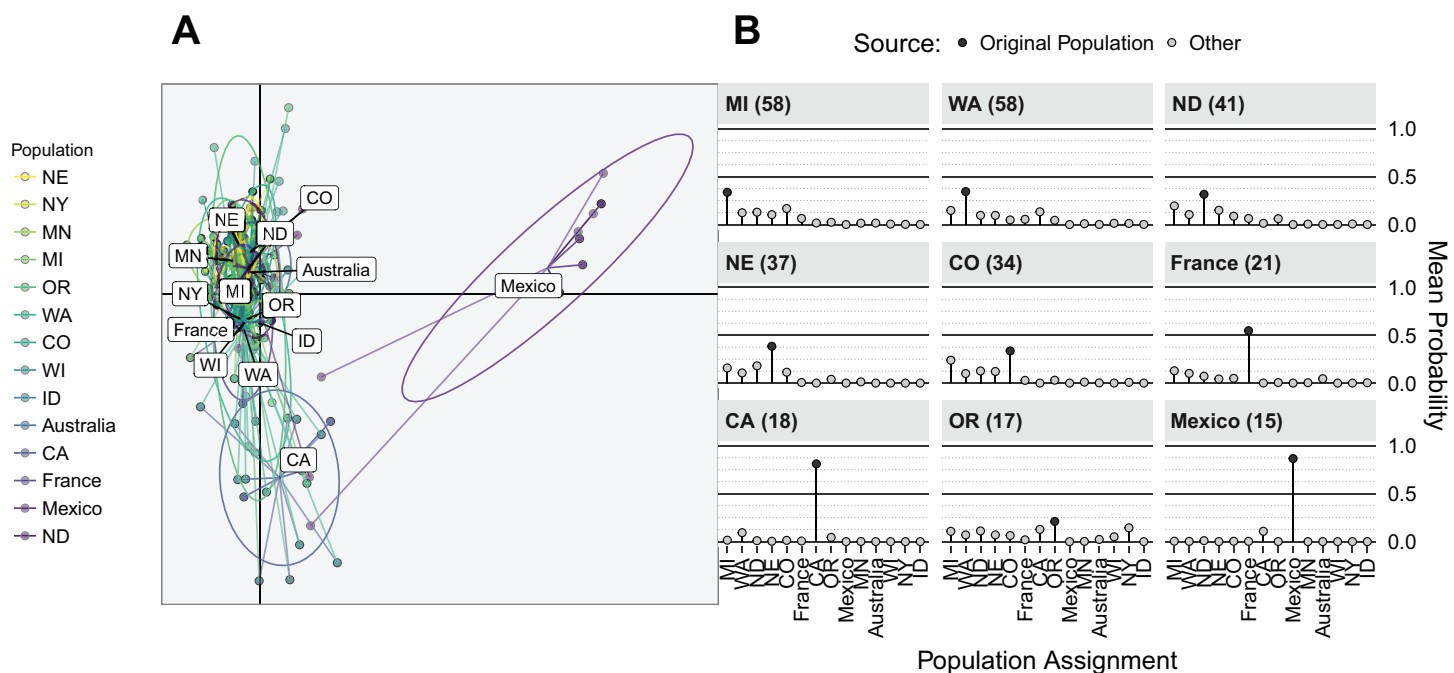

**Figure 5** **Discriminant analysis of principal components (DAPC) on regions showing that Mexico is differentiated from other populations.** (A) Scatter plot of first two components from DAPC. Points represent observed individuals connected to the population centroids with ellipses representing a 66% confidence interval for a normal distribution. The center of each component is represented as black grid lines. (B) Mean population assignment probability from the DAPC for all populations with $N > 10$ (facets). Populations represented along the horizontal axis and probability of assignment on the vertical. Numbers next to source populations indicate population size. All values sum to one.

### Discriminant analysis of principal components

Discriminant analysis of principal components was performed by grouping Region with the first 21 principal components, representing 88.1% of the total variance. The first discriminant axis (representing 63.9% of the discriminatory power) separated the centroid for the Mexico isolates from the rest of the data, indicating strong differentiation (Fig. 5B). The second discriminant axis, representing 10.8% of the discriminatory power, separated the centroid for the CA isolates. The mean population assignment probabilities for all populations with $n > 10$ showed that only isolates from Mexico, CA, and France had >50% probabilities of being reassigned to their source populations (Fig. 5A).

Discriminant analysis of principal components grouping by cultivar used the first 20 principal components, representing 89% of the total variance. The first two discriminant axes (representing 100% of the discriminatory power) failed to separate any of the cultivars where the mean posterior assignment probabilities were 34% (G122), 35.9% (Beryl), and 30.1% (Bunsi). DAPC grouping by Region and Year used the first 15 principal components, representing 80.3% of the total variance. The North Central United States populations (NE, MI, CO, ND) did not appear to have any variation across time in contrast to WA, which showed a shift in population structure in the last year of sampling, 2008 (Fig. 6). Further analysis of this population revealed that all 12 isolates in WA circa

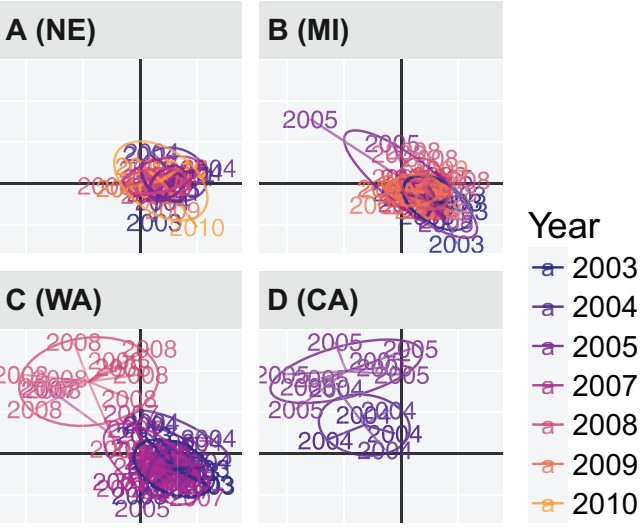

**Figure 6 Scatter plot of discriminant analysis of principal components (DAPC) on Regions and Years showing non-differentiated temporal variation NE and MI and temporal variation in WA and CA.** Points (text labels) represent observed individuals connected to the population centroids with ellipses representing a 66% confidence interval for a normal distribution. The center of each component is represented as black grid lines. A more detailed view is shown in Fig. S5.

2008 originated in a wmn; nine haplotypes were shared with CA, and three were shared with France (Fig. 4; Fig. S4).

## DISCUSSION

In this study, we characterized the diversity of *S. sclerotiorum* from dry bean fields across the United States. Our results suggest that, broadly, populations from wmn reflect the populations of the surrounding regions, indicating that resistance screening may be successful within regions. We found significant population differentiation by geographic region and year, mainly differentiated into three broad North American groups based on shared haplotypes and posterior groupings, a Coastal Region, Midwestern Region, and Mexico. To date, with 366 isolates, this is the largest single population genetic study of *S. sclerotiorum* assessing population structure within managed and unmanaged agricultural environments. These findings indicate that the wmn can be effective at screening for potential resistant lines within growing regions.

We found that the best predictors of genetic structure are Region and Year, supporting the hypothesis that *S. sclerotiorum* populations are spatially structured (*Carbone & Kohn, 2001*). Borrowing a technique often used in the ecological literature, we used dbRDA to elucidate the effect of all variables (MCG, Region, Source, Year, Host, and Aggressiveness) (*Legendre & Anderson, 1999*). From the initial results, it appeared that the most important factors for predicting genetic structure were MCG, region, and year. When we inspected the biplot of the initial results, we saw that the most important predictors were "MCG 44," "MCG 5," and "MCG 9." We believe that this was driven by the fact that these particular MCGs have uneven MLH distributions, meaning that they are heavily associated with one

particular MLH (Fig. 1). We note these results with caution because of the apparent multicolinearity between MCG and Region, which is a violation of the analysis (*Legendre & Anderson, 1999*). While the results indicated that Mexico and the year 2005 were the two most important variables, it's worth noting that all Mexico isolates were collected in 2005 (Fig. 2). The results also show that the Vista cultivar explains some of the variance, but this represents six isolates in MI, and thus we cannot draw broad conclusions from this axis. Aggressiveness and source field had little to no effect on prediction of genetic diversity. These results are in agreement with studies that examined differentiation based on Host (*Aldrich-Wolfe, Travers & Nelson, 2015*) and Aggressiveness (*Atallah et al., 2004*; *Attanayake et al., 2012, 2013*) reporting little or no correlation of genetic diversity to these variables. This indicates that (a) breeders should keep in mind regional differences when assessing resistance and (b) it is possible that we have not yet measured biologically relevant variables that can predict genetic differentiation, which could include variables such as soil community composition.

While aggressiveness was not shown to predict genetic structure, it is an important factor in breeding efforts, and we observed significant differences in aggressiveness based on Region (Fig. 3; Table S2). These results show a similar pattern to what was found previously in *Otto-Hanson et al. (2011)* with the exception of North Dakota, which increased in mean aggressiveness from 5 to 5.77. This increase was due in part to new data from producer field isolates collected after the previous study. These straw tests were performed by a different person for these later isolates, which could suggest a more lenient or strict scoring system. However, when we examined the within-region differences, we found no significant effect by individual. Many of the ND isolates fell within the 6–7 range, which denotes a physical boundary (disease symptoms around the second node) between intermediate and susceptible (*Otto-Hanson et al., 2011*). Thus, we observed a shift in aggressiveness without a significant shift in genotypic structure, which may indicate that aggressiveness may be controlled by environmental factors as opposed to genetic profile.

The primary interest of this study was to assess if isolates sampled from wmn represent isolates from producer fields within the region (*Steadman, Eskridge & Powers, 2003*; *Otto-Hanson et al., 2011*). According to our AMOVA results, we have evidence for differentiation at the Region and Year, but little to no differentiation between wmn isolates and production field isolates (Table 4). This lack of differentiation, however, may reflect the breeder practice of inoculating screening plots with sclerotia collected from sources within the region. When we analyze the AMOVA results in light of the DAPC results (Fig. 5), it becomes clear that the regional patterns of differentiation are largely driven by isolates from Mexico and CA. Isolates from these Regions had a higher posterior probability (>0.75) of being reassigned to their own populations than any other (Fig. 5A). All other populations in comparison (except France) has reassignment probabilities of <0.5, which is reflected in the failure of the first two discriminant functions to separate these populations (Fig. 5B).

Despite the evidence that Mexico and CA contributed to much of the population differentiation, Regions like WA still had a large amount of internal variation. The two

distinct clusters for the WA Region showed that the 2008 population appeared differentiated and, under further investigation, we found that all the haplotypes from this year were shared between CA and France (Figs. 4 and 6; Fig. S5). All of the isolates from WA in 2003–2005, and 2008 came from the same wmn; within the wmn, those in 2003–2005 came a Northeastern field location cropped with dry bean since 2002, and those in 2008 from a Southeastern field that was previously cropped with brassica, sundgrass, peas, beans, and potatoes (P. Miklas, 2017, personal communication). Both of these fields were inoculated with sclerotia in 2002, the Northeastern field with sclerotia provided by a commercial bean producer and the Southeastern field with sclerotia from peas (although this was thought to be unsuccessful). Despite this information, it is still unclear what has contributed to the differentiation of the 2008 population from WA or why it shares haplotypes with CA and France. When we assessed aggressiveness between the two fields across years with an ANOVA model, we found that there was a slight effect based on field ($P = 0.0127$). While the evidence may suggest host as being a factor, previous studies have shown no significant differentiation across host species (*Aldrich-Wolfe, Travers & Nelson, 2015*). It was of interest to compare our data with that of *Aldrich-Wolfe, Travers & Nelson (2015)*, but we found that, due to differences in data generation, we were unable to confidently perform a comparison (see supplemental file compare-aldrich-wolfe.md, https://github.com/everhartlab/sclerotinia-366/blob/master/results/compare-aldrich-wolfe.md; *Kamvar et al., 2017*).

With the exception of the WA Region, populations that were sampled across several years appeared to be relatively stable over time with overlapping distributions in the DAPC (i.e., NE and MI, Fig. 6). DAPC is based on the principal components of allele counts (*Jombart, Devillard & Balloux, 2010*). Unlike Bruvo's distance, this does not take into account the magnitude of the difference between alleles, which could inflate the distance measure in the presence of private alleles (*Bruvo et al., 2004*). While we found no evidence of private alleles in the Mexico and CA isolates, we did find that the alleles driving the first axis in Fig. 5A (alleles 174, 256, and 372 in loci 7-2, 8-3, and 9-2, respectively) were over-represented in Mexico (where >75% of the alleles came from the region). However, all three of these alleles, (i) conform to the expected stepwise mutation model (*Bruvo et al., 2004*) and (ii) are at or near the extremes of the total range (except for allele 372 at locus 9-2). Moreover, the fact that we find three alleles at three independent loci segregating the Mexican genotypes suggests that the pattern separating these populations from the others was not an artifact. We believe that the differences in populations observed from Mexico may be due to differences in climate that allow greater diversification via sexual outcrossing.

Many of the isolates in our study were from temperate climates and the only isolates representing a sub-tropical climate were from Mexico. It has been proposed within the *S. sclerotiorum* literature that isolates from sub-tropical and tropical climates are differentiated or more variable than populations from temperate climates (*Carbone & Kohn, 2001*; *Attanayake et al., 2013*; *Lehner & Mizubuti, 2017*). This has been attributed to the notion that the fungus has the chance to undergo more reproductive cycles in the warmer climate (*Carbone & Kohn, 2001*; *Attanayake et al., 2013*). The strongest evidence
to date supporting this hypothesis is from *Attanayake et al. (2013)*, showing that populations in sub-tropical regions of China have been found to be more variable, sexually reproducing, and unrelated to populations in temperate regions of the United States. This result however, may be driven more by geography and agricultural practice as opposed to climate.

The results from our shared haplotype analysis showed several populations with at least one haplotype between them, except for Mexico and two states that had fewer than three samples each (Fig. 4). Our network-based approach by treating the haplotypes as edges and weighting each edge with the inverse of $P_{sex}$ treated the edges as springs connecting the populations with the strength proportional to the probability of obtaining the same haplotype as a clone. This allowed us to use a graph walking algorithm to see how close the populations were, simply based off of the proportion of clones they shared. The most abundant haplotype was shared across four populations, but its high value of $P_{sex}$ meant that it did not contribute significantly to the overall structure. The graph walking algorithm was able to divide the network into three groups, but had a modularity of 0.17, which indicates that the groups are only weakly differentiated.

The widespread nature of MLH in both wmn and production fields with relatively small values of $P_{sex}$ may indicate the spread of inoculum between regions. While seed-borne transmission is thought to be of insignificant epidemiological importance (*Strausbaugh & Forster, 2003*), it has since been shown that *S. sclerotiorum* infections can be transmitted through seed (*Botelho et al., 2013*). Thus, we hypothesize that shared haplotypes between populations may arise due to transmission events of seed or sclerotia. This could explain the fact that we see shared haplotypes with low $P_{sex}$ values shared between Australia, France, and the United States. While we speculate that these transmission events are rare due to the genetic structuring by Region, these results suggest that seed-borne infections may indeed reflect a source of inoculum. This may, in turn increase the risk of introducing new sources of genetic variation through potential outcrossing events.

When we tested for sexual reproduction, we were unable to find evidence for it in any region except for Australia and CA. While the Australia population had a non-significant value of $\bar{r}_d$—which would suggest that we cannot reject the null hypothesis of random mating—the sample size was insufficient from which to draw conclusions (*Milgroom, 1996*; *Agapow & Burt, 2001*). The low value of $\bar{r}_d$ in the CA population may represent sexual reproduction, but we can see in Fig. 6 that there is differentiation by year. Thus, this could also be an artifact of sampling two different populations, which is known to reduce the value of $\bar{r}_d$ (*Prugnolle & de Meeus, 2010*).

The previous study of the wmn populations used MCGs to assess genotypic diversity (*Otto-Hanson et al., 2011*). Historically, MCGs have been used as a proxy for clonal lineages, and thus, of interest in this study was testing the association between MLHs and MCGs (*Kohn, Carbone & Anderson, 1990*; *Leslie, 1993*; *Kohn, 1995*; *Carbone, Anderson & Kohn, 1999*; *Schafer & Kohn, 2006*; *Otto-Hanson et al., 2011*). Our results, however, do not support this assumption. It can be seen in Fig. 1A that the most abundant MCG contains

several MLHs, but the diversity of those MLHs are low as indicated by the evenness (transparency), which indicates that there is one dominant MLH ("MLH 25"). What is not shown in Fig. 1A is the MLHs that are shared between MCGs. This is illustrated in both Table 3 and Figs. 1B and 1C. It could be argued, however that "MLH 25," with its high value of $P_{sex}$ represents different true MLHs across the five MCGs it occupies, but this does not account for the overall structure of Fig. S3 where, for example, "MLH 75" ($P_{sex} = 1.81e^{-4}$) is compatible with 57 other haplotypes through three MCG when the population structure of *S. sclerotiorum* is known to be clonal.

Over the past few years, researchers have noticed inconsistencies among the relationship between MCGs and MLHs (*Carbone, Anderson & Kohn, 1999*; *Attanayake et al., 2012*; *Aldrich-Wolfe, Travers & Nelson, 2015*; *Lehner et al., 2015*). Either several MCGs belong to one MLH, which could be explained by insufficient sampling of loci; several MLHs belong to one MCG, which could be explained by clonal expansion; or a mixture of both. Some studies have shown a correlation between MCG and MLH (*Carbone, Anderson & Kohn, 1999*; *Aldrich-Wolfe, Travers & Nelson, 2015*; *Lehner et al., 2015*), whereas other studies have shown no apparent correlation, even on small spatial scales (*Atallah et al., 2004*; *Attanayake et al., 2012, 2013*).

One long-held assumption was that MCGs (as determined via barrage reaction) represent vegetative compatibility groups (VCGs) (*Kohn, Carbone & Anderson, 1990*; *Schafer & Kohn, 2006*; *Lehner et al., 2015*), which are known to have a genetic component (*Saupe, 2000*; *Hall et al., 2010*; *Strom & Bushley, 2016*). While our protocol for assessing MCGs utilized DS Medium (*Cubeta, Sermons & Cody, 2001*) as compared to Patterson's Medium or PDA (*Schafer & Kohn, 2006*) for the MCG reactions, the patterns we observe are not dissimilar from what have previously been reported in the literature. It has been demonstrated in several Ascomycetes—including *Neurospora crassa* (*Micali & Smith, 2003*), *S. homoeocarpa* (*Jo et al., 2008*), *Verticillium dahliae* (*Papaioannou & Typas, 2014*), and *S. sclerotiorum* (*Ford et al., 1995*)—that barrage reactions are independent from stable anastomosis. Thus, the inconsistencies in this study and other studies indicate that researchers studying *S. sclerotiorum* should not rely on MCG data derived from barrage reactions as an indicator for genetic diversity.

### Limitations

One of the main limitations of this study is the focus on *P. vulgaris* as a host. It has been shown that *S. sclerotiorum* in the Midwestern United States does not have a particular preference for host (*Aldrich-Wolfe, Travers & Nelson, 2015*). If the distribution of *S. sclerotiorum* is even across agricultural hosts in the United States, then our sample may yet be representative of the genetic pool present in other crops and weedy species. Additionally, while we found no significant association between genotype and aggressiveness, it is important to note that the straw test is only one measure of aggressiveness. Additional phenotypes for aggressiveness should be evaluated for future studies.

Another limitation was the microsatellite markers used for this particular study (*Sirjusingh & Kohn, 2001*). The haplotype accumulation curve showed no indication of a plateau, indicating that if we had sampled more loci, we would have resolved more MLHs.

While 16 loci showed us similar results and began to show a plateau for the haplotype accumulation curve, we were unable to use these results due to our uncertainty in the allele calls for these five extra loci. With the availability of an optically mapped genome (*Derbyshire et al., 2017*), future studies describing the genetic diversity of *S. sclerotiorum* should employ techniques such as Genotyping-By-Sequencing (*Davey et al., 2011*), Sequence Capture (*Grover, Salmon & Wendel, 2012*), or Whole Genome Sequencing.

## CONCLUSION

This study represents the largest genetic analysis of *S. sclerotiorum* from the United States to date, giving us a unique insight to continent-wide population structure and relationships between phenotypic and genotypic variables. Populations in wmn appear to show no significant differentiation when compared to their production field counterparts, suggesting that the wmn populations of *S. sclerotiorum* may be considered representative of the surrounding regions. While we found no direct relationship between haplotype and severity, it is evident that there is a gradient of severity by region, further supporting the need for screening in multiple locations. Based on our analysis of the relationships between MCG and MLH, we found no clear evidence that the two are directly related, suggesting that MCG does not necessarily represent vegetative compatibility groups and thus should not be used as a proxy for identifying clones.

## ACKNOWLEDGEMENTS

The authors would like to thank Rebecca Higgins for technical support in generating the data for the MCG assessment, aggressiveness ratings, and genotyping; and for providing valuable insights into the historical context of the data collection and curation. We would also like to thank Denita Hadziabdic and two other anonymous reviewers for their valuable comments and insights that improved the quality of the manuscript.

### Funding

Funding for this research was provided by a Layman Award (#2446) to Sydney E. Everhart, USDA-ARS National Sclerotinia Initiative (#58-5442-2-209) to James R. Steadman and Sydney E. Everhart, and start-up funds from the University of Nebraska-Lincoln to Sydney E. Everhart. The funders had no role in study design, data collection and analysis, decision to publish, or preparation of the manuscript.

### Grant Disclosures

The following grant information was disclosed by the authors:
Layman Award: #2446.
USDA-ARS National Sclerotinia Initiative: #58-5442-2-209.

### Competing Interests

The authors declare that they have no competing interests.

## Author Contributions

- Zhian N. Kamvar analyzed the data, contributed reagents/materials/analysis tools, wrote the paper, prepared figures and/or tables, reviewed drafts of the paper.
- B. Sajeewa Amaradasa analyzed the data, contributed reagents/materials/analysis tools, wrote the paper, reviewed drafts of the paper.
- Rachana Jhala performed the experiments, contributed reagents/materials/analysis tools, reviewed drafts of the paper.
- Serena McCoy performed the experiments, contributed reagents/materials/analysis tools, reviewed drafts of the paper.
- James R. Steadman conceived and designed the experiments, contributed reagents/materials/analysis tools, reviewed drafts of the paper.
- Sydney E. Everhart analyzed the data, contributed reagents/materials/analysis tools, wrote the paper, reviewed drafts of the paper.

## Data Availability

All scripts, data, and resources used to generate the results presented in this publication (including Supplementary Information) are fully reproducible and available at the Open Science Framework: https://osf.io/ejb5y (*Kamvar et al., 2017*).

## Supplemental Information

Supplemental information for this article can be found online at http://dx.doi.org/10.7717/peerj.4152#supplemental-information.

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
