# Peer review of "Population structure and phenotypic variation of Sclerotinia sclerotiorum from dry bean (Phaseolus vulgaris) in the United States"

_PeerJ, doi:10.7717/peerj.4152_

## Round 0.1 · original submission · Minor Revisions

Your manuscript has been seen by three qualified reviewers. Based on their detailed assessments and my own, I feel this work is well suited for publication in PeerJ after a number of minor revisions.

Reviewer 1 ·

Basic reporting

-The article is well written.
-Literature References are relevant and fits the study and within a broader field.
-The structure of the article is of good standard. The format of PeerJ is the opportunity to provide greater detail and more in-depth discussions that may be restricted in other journals. However, there is a delicate balance of introducing as much detail as necessary while retaining the attention of the reader. This article is on the borderline of introducing too much detail. For instance, is the use of Figures 1, 2 and 4 all necessary? The underlying question is, Do the Figures and additional detail in the paper enhance the paper or create a distraction from the primary message of the study? And while there is great detail on different analyses there is lack of depth on the Mexican population? Is this an important population and why?
-Complete study without indications of fracturing of the research to increase publication count.

Experimental design

-Well defined gap in scientific knowledge, confirms the efficacy of white mold screening nurseries through population genetics analysis and associated aggressiveness of isolates.
-Appropriate use of technologies for investigating genetic populations with that are predominantly clonal. Identified and evaluated an appropriate phenotype of interest (aggressiveness). Identified the limitations of the study, in that the pathogen is not host specific and may influence conclusions of the study.
-Methods would allow for an investigator to replicate the same or similar study to confirm results or develop new studies.

Validity of the findings

-Study provides novel results in the scientific field of S. sclerotiorum on dry beans an important agricultural commodity. Potentially validates the continued use of nurseries for resistance selection in breeding and introduces a potential area of concern/focus for future research with the Mexican population of pathogens.
-Data is provided in repository: https://github.com/everhartlab/sclerotinia-366/. Review of data available (includes: year, location, host…) indicates the opportunity for evaluation/use of data to confirm results and/or in future studies. Statistical analysis of data was appropriate.
-Conclusions are well stated and appropriate.

Additional comments

Overall this paper did a wonderful synthesis of the topic while applying and interpreting the results with good scientific standards. I found only one important area of correction which was in the abstract requiring the clarification of "11 states in the United States of America,...". This is noted in the material and methods but is absent in the abstract.

From a scientific perspective, it should be noted that additional phenotypes for aggressiveness should be evaluated in future research. The straw test is only an indicator for one type of pathogen potential and limits interpretation of the populations. The conclusion given in this study is valid but greater resolution may have been possible with more phenotype data (and of course larger populations). I was also interested in more interpretation/analysis as to why the Mexican population did not have MLH shared with any other regions, the absence of clarity on this is done at a loss.

·

Basic reporting

Manuscript is written clearly, with appropriate references but limited introduction. Please see general comments reagrding that. The authors has clearly stated research questions, hypothesis, detailed M&M, concise results, and extensive discussion.

Experimental design

Research questions were clear and addressed appropriately in subsequent analyses. Methodology and data analyses are described with sufficient details to allow reproducibility.

Validity of the findings

Data was robust, available for reproducibility and well explained.

Additional comments

Population structure and phenotypic variation of Sclerotinia sclerotiorum from dry bean in the United States by Kamvar et al. utilized microsatellite loci to answer number of interesting questions including evaluating phenotypic and genetic diversity of S. sclerotiorum in the nurseries (here referred as natural populations since no control was used to limit disease spread) using regional differences and across different time intervals (span of nine years). In addition, the authors investigated correlation between mycelial compatibility groups and multilocus haplotypes among these populations. Introduction is a bit short and I would suggest expanding few sections (please see specific comments below). Materials and methods are precise and well written and I really appreciated data availability, including all sorts of analyses, which was refreshing. I also like the acknowledgment of shortcomings of the analyses (compound microsats example), which can be challenging to work with. Results were explained well with few exceptions that need some clarification. Overall, well written manuscript with interesting and relevant results for nursery producers/growers. As such, I recommend it for publication with minor revisions.

Reviewer 3 ·

Basic reporting

The paper focuses on addressing a question of the genetic diversity of Sclerotinia and addressing how this genetic diversity could be ligated to virulence and compatibility between strains. The context provided for the paper is sufficient to state the goal of the research, and the authors show knowledge of the system and the analyses required to achieve the stated goals. The article is sound and data/code for analyses have been made public and easily accessible. In general, the conclusions were supported by the data, there are some points that required some clarification, but overall the research was well developed and the data is nicely presented to follow up the document.

Experimental design

The paper is one of the most extensive population studies in plant pathogens addressing questions on the structure of the population and the correlation of genetic diversity with specific traits. The goals and research question stated by the authors were mostly addressed with the data and there are points that despite the difficulty of the question, there are good approximations to the answer. For instance, the limited information provided by SSRs could not potentially lead correlation with phenotypic traits, therefore the authors acknowledge this, and approach the question using different statistical methods.
The methods and the analyses conducted are well explained and the authors provided all the code and data for corroborating the results.

Validity of the findings

Overall the study provides an extensive view of the genetic diversity of S. sclerotiorum in screening nurseries and commercial fields of dry bean. Despite that the question of how genetic diversity links phenotypic traits like MCGs and virulence has been approached before, the authors analyzed a large number of isolates using multiple markers and different statistical approaches to answer this question. The result is still negative since there are limitations by the markers, and since this pathogen has reduced diversity. Sclerotinia is soilborne pathogen and it is expected that there should be constraint populations at the geographical level, however, there is a reduced diversity suggesting little differentiation among regions. This is addressed by the authors, where soil or contaminated plant material could have played an important role on the transmission of this pathogen. In addition, the goal of establishing a census of the genetic diversity and its relation to aggressiveness is major task but necessary to establish a baseline for breeders to target a representative pool of the pathogen’s population.
Nevertheless, the authors go through a good job of addressing the issues and limitations of the study. There are some points or comments that I would recommend to the authors to discuss and/or consider, those were included on the general comments.

Additional comments

• The availability of the data and the analyses posted on github was really helpful and it made very enjoyable to read the paper and understand some of the logic of the authors, it has been a great experience. It also provides a good view on the paper and it helps to assess paper and give recommendations on the paper. Kudos! I want to compliment the authors for making these resources available.

• Since populations for certain areas were only collected within a single year, variation between years and region should also be looked at with caution. Are year and region still important if samples with more than one year are retained? How much variability is explained if so? It will a good way to corroborate if there is a continuum of genotypes or every year is bottleneck increasing diversity and to determine how much populations sampled once contribute to the analysis. However, the authors are aware of this on line 350-352.

• One thing to be addressed is how well these microsatellites represent the whole genome, this was not addressed on Sirjusingh and Kohn (2001) maybe due to the lack of the genome sequence. However, this could explain the lack of power of the existing set to represent the haplotypes in the population. Despite, that most studies are using reduced genome approaches or more powerful techniques, it will be informative for other researchers still using this set of microsatellites to have this information. Njambere et al. (2010; 10.1139/G10-019) did an approximation for S. trifoliorum using linkage groups, but I am not aware of something similar for S. sclerotiorum to corroborate these SSRs.

• The relation between MCGs, MLH and aggressiveness is quite interesting, however, it is hard to follow in the text. The graph in github (https://github.com/everhartlab/sclerotinia-366/blob/master/results/mlg-mcg.md), summarizes really well some on this information as well as table 3. Maybe you can consider including this graph either on the article or add a column to table 3 with the average aggressiveness. The graph might be more meaningful since you can see the variability of aggressiveness of the different strains by the different factors.

• L384-404 The authors have a discussion on differentiation of the population based on region. Then, the discussion is centered on how regions like WA still had a considerable amount of variation within the US locations sampled. However, one of the points discussed is that one of the locations was inoculated in 2002 and then crop history differed between the two locations sampled. As the authors suggested there is little or minimal differentiation between isolates from different hosts. Nonetheless, the source of the sclerotia used to inoculated the fields is also different. This could be also part of the differences that the authors see in 2008. Despite that other studies have indicated a limited differentiation between hosts, it seems that there is some effect on the genetic. Is the virulence different on these isolates with respect to other isolates? Aldrich-Wolfe et al. (2015) presents information on the allele sizes for the markers used, are these haplotypes present in the current study? It will be interesting to determine if most haplotypes are share or not, and if those present across multiple hosts have a different virulence. However, the major point of the paper is dry bean but the history on crop rotation could explain some of the variability across years.

• P6L262 varaibles change to variables

---

## Round 0.2 · accepted · Accept

Your revised manuscript has addressed the previous reviewer concerns and I feel it is well suited for publication in PeerJ.